# A Simplified Method of Iceberg Hydrodynamic Parameter Prediction

Dmitry Nikushchenko [1,*], Anton Stepin [1], Ekaterina Nikitina [1], Nikita Tryaskin [2], Alexander Makovsky [2], Konstantin Kornishin [3] and Yaroslav Efimov [3]

1 Institute of Hydrodynamics and Control Processes, Saint-Petersburg State Marine Technical University, Lotsmanskaya St., 3, 190121 Saint-Petersburg, Russia; astiopin@gmail.com (A.S.); nikitina@corp.smtu.ru (E.N.)
2 Laboratory of Applied Hydrodynamics, Saint-Petersburg State Marine Technical University, Lotsmanskaya St., 3, 190121 Saint-Petersburg, Russia
3 Department of Offshore Oil and Gas Field Development, Gubkin Russian State University of Oil and Gas (National Research University), 65, Leninsky Prospect, 119991 Moscow, Russia; kornishink88@yandex.ru (K.K.); efimov.yo@gmail.com (Y.E.)
* Correspondence: dmitry@nikushchenko.ru

**Abstract:** The present article is devoted to the safe operation of oil platforms in difficult ice conditions. To ensure the safety of offshore facilities, it is essential to deviate a drifting iceberg's trajectory that may lead to an emergency with the help of available technological means. To reliably predict the behaviour of icebergs when they are towed or deviate from their previous course, it is necessary to determine the hydrodynamic and aerodynamic characteristics of the iceberg. This paper proposes a simplified method for determining the hydrodynamic and aerodynamic characteristics of an iceberg. Key concept of the proposed approach include replacing the iceberg waterline with an equivalent ellipse. This diminishes and then shifts the hydrodynamic characteristic determination of the iceberg's underwater section to determining the resistance of a three axial ellipsoid or elliptical cylinder depending on the shape of the iceberg's underwater section. The hydrodynamic characteristics of several real icebergs determined by the proposed method are compared with the results of their numerical simulation using the CFD approach. The proposed approach provides a quick assessment of the hydrodynamic characteristics of icebergs when information on its underwater section is unreliable, inconsistent or absent.

**Keywords:** iceberg towing; ice management; iceberg resistance; CFD; hydrodynamic parameters; numerical simulation; towing simulation

## 1. Introduction

Iceberg towing is a common marine operation carried out in a number of offshore exploration and production projects on the shelf of the North Atlantic and Arctic oceans to avoid collision with offshore structures. The motion of an iceberg in the process of towing, as the motion of any material body under the action of applied loads obeys the basic laws of mechanics. However, mathematical modelling of towing has some features associated with:

- irregular shape of the above-water and underwater parts of icebergs, determination of which is complicated in real environment conditions;
- significant displacement of the iceberg compared to the vessel towing it;
- influence of various environmental factors (wind, currents, sea waves, etc.).

In order to understand the mechanics of iceberg towing, it is necessary to conduct both full-scale iceberg towing simulations, and numerical and basin modelling. Each of these methods has its own advantages and disadvantages. Full-scale simulations make it possible to obtain real data on the behaviour of the "vessel-rope-iceberg" system. Yet, they are very costly and do not provide a quantitative assessment of the influence of each factor that determines the motion of an iceberg. Mathematical modelling and computer

simulation allows evaluating required aerodynamic and hydrodynamic characteristics of the iceberg. However, they are limited by the scope of the simplified iceberg towing model, which usually omits some important phenomena that are typical for real iceberg towing. Iceberg towing operations are described in many scientific publications (for example, [1]), some research has also been carried out on modelling the motion of an iceberg during towing (for example, [2]). C-CORE has also studied icebergs for the ice management of offshore exploration and production units (for example, [3–5]).

The first documented case of iceberg towing as part of an offshore mining project dates back to April, 1970, on request by the Tenneco Oil & Minerals Ltd. Simulation work was carried out to determine the possibility of protecting the first exploration well "Leif" P-38 on the shelf of the Labrador Peninsula, which was drilled by the moored "Typhoon" vessel [6]. CCS Dawson, a Canadian coast guard vessel, carried out thirteen tows of three icebergs weighing up to 1000 tons during 1971–1973 [1]. A net was used as a towing system. The towing time was approximately 15–30 min and the maximum achieved towing speed did not exceed 1 m/s. The obtained simulation data demonstrated that the average drag coefficient was 1.2 for iceberg fragments. This value is in good agreement with the drag coefficient of an ordinary cylinder, obtained during several laboratory studies on the flow around it at the corresponding values of the Reynolds number.

Attempts to develop a numerical model of iceberg motion in open water were made by G. Murphy [7], as the possibility was considered to conduct a numerical simulation of changes in the towing force and mass of an iceberg. A series of small-scale simulations to determine the drag force of icebergs of various shapes was carried out by F. Mauviel in 1980 [8]. A simulation of the motion od an iceberg during towing was carried out in a number of studies by A. V. Marchenko [2,9]. For the first time, 3D models of icebergs were created as part of the Terra Nova project [10] to evaluate the possible damage to offshore oil and gas facilities. The need to calculate the shape of icebergs was studied by various researchers [11–13].

Currently, there are no comprehensive studies on iceberg towing covering its major stages: *iceberg morphometry* (measurements of above- and underwater parts with UAV and an echo sounder for the 3D model), *iceberg towing* (during which all kinematic and dynamic parameters of its motion and additional environmental characteristics are recorded), and then *towing tank and CFD modelling* of hydrodynamic and aerodynamic characteristics of the considered iceberg. In this study, the authors attempt to fill this gap. The study was based on the results of iceberg towing simulations in 2017, as well as the processing of the results of those simulations. A detailed description of the iceberg towing trials is given in [14]; hydrodynamic and aerodynamic characteristics of the studied icebergs are discussed in [15,16].

Section 2 of the present article introduces the new simplified approach to iceberg hydrodynamic parameter prediction on a basis of their representation by equivalent icebergs (Section 2.1) and an approximation by a polynomial representation of CFD simulations (Sections 2.2 and 2.3). The suggested approximations are described in the (Section 2.4). In Section 3 we compare the results of the suggested approach with the simulation measurements of models of four real icebergs.

## 2. Simplified Approach to Iceberg Hydrodynamic Parameter Prediction

Prediction of the hydrodynamic characteristics for the underwater part of an iceberg is crucial when dealing with the following practical applications:

- designation of a vessel with sufficient shaft thrust and suitable towing equipment for the specific world ocean sailing area;
- planning and carrying out a change in the drift trajectory of an iceberg with known surface shape;
- estimation of the iceberg's stability during towing;
- minimizing the duration of the transitional processes during towing [17];

- determination of the stress state parameters of the towing system under various environmental conditions, including the evaluation of oscillatory processes [18].

The huge variety of sizes and shapes of drifting icebergs does not allow for creating a sufficiently correct mathematical model of the towing order motion, which includes an icebreaker—towing vehicle, a towing system and the object to be towed.

Primary simplification concerns the shape of the considered icebergs. Information on the surface shape of the iceberg can be obtained, for example, involving aviation, namely using aerial photography. The volume of the underwater part of the iceberg is known to be approximately six times greater than the volume of ice located above water. Specifically, this ratio allows us to estimate the volume of the underwater part of the iceberg.

In addition, the results of aerial photography helps calculate the waterplane area and the average height of the above-water part of the iceberg. Therefore, this allows us to estimate the volume of the entire iceberg and, accordingly, its mass. Then we can calculate the second moment of area of the waterplane, metacentric radius and metacentric height. The aforementioned parameters are mandatory for assessing the static roll during towing and oscillation in waves of varying intensity.

For the subsequent calculations, the waterline can be approximated by analytical expressions in the first approximation. For example, it can be a circle or an ellipse. The areas of those figures should be as close as possible to the waterplane area of the towed iceberg.

*2.1. Equivalent Icebergs*

The considered approach is based on a substitution of any random iceberg with an equivalent semi-ellipsoid or elliptical cylinder (depending on the underwater shape of the iceberg under consideration). This creates the equivalent ellipse over the actual waterline of the iceberg as shown in Figure 1, where $L$ is the maximal length of the iceberg and $C$ is the width in middle area. As a result, the semi-axis of the equivalent ellipse are $a = \frac{L}{2}$ and $b = \frac{C}{2}$. In this case, let us assume that the longitudinal axis $x$ of the reference frame is directed along the large ellipse axis, and the transversal one $y$ is directed along the small ellipse axis. The reference frame origin is situated in the geometrical centre of the ellipse.

In the present article, we distinguish the total forces into three groups: static, inertial and hydrodynamic. Let us suppose that the considered iceberg is in an equilibrium state, i.e., its weight and flotation force do not influence the iceberg's motion.

Hydrodynamic force coefficients can be written in the following form:

$$C_x = \frac{R_x}{\frac{\rho U^2}{2} V^{\frac{2}{3}}}, \quad C_y = \frac{R_y}{\frac{\rho U^2}{2} V^{\frac{2}{3}}}, \tag{1}$$

where $R_x$ and $R_y$ are the longitudinal and transversal hydrodynamic forces, accordingly, $\rho$ is the water density, $U$ is the speed of the iceberg relative to the water, and $V$ is the iceberg's underwater volume. Here, we assume that the characteristic area is $V^{\frac{2}{3}}$ and the characteristic length is $V^{\frac{1}{3}}$. Hereafter, we assume that the longitudinal and transversal hydrodynamic forces are functions of the drift angle $\beta$ only.

In this case, the total drag is as follows:

$$C_d = \sqrt{C_x^2 + C_y^2}. \tag{2}$$

The turning moment coefficient may be written using the superposition principle in the following form:

$$M_z(\beta, \omega) = M_z(\beta) + M_z(\omega), \tag{3}$$

where $M_z(\beta)$ and $M_z(\omega)$ are the positional and rotational parts of the moment, respectively.

The positional moment coefficient is as follows:

$$C_{m_z}(\beta) = \frac{M_z(\beta)}{\frac{\rho U^2}{2}V}.\tag{4}$$

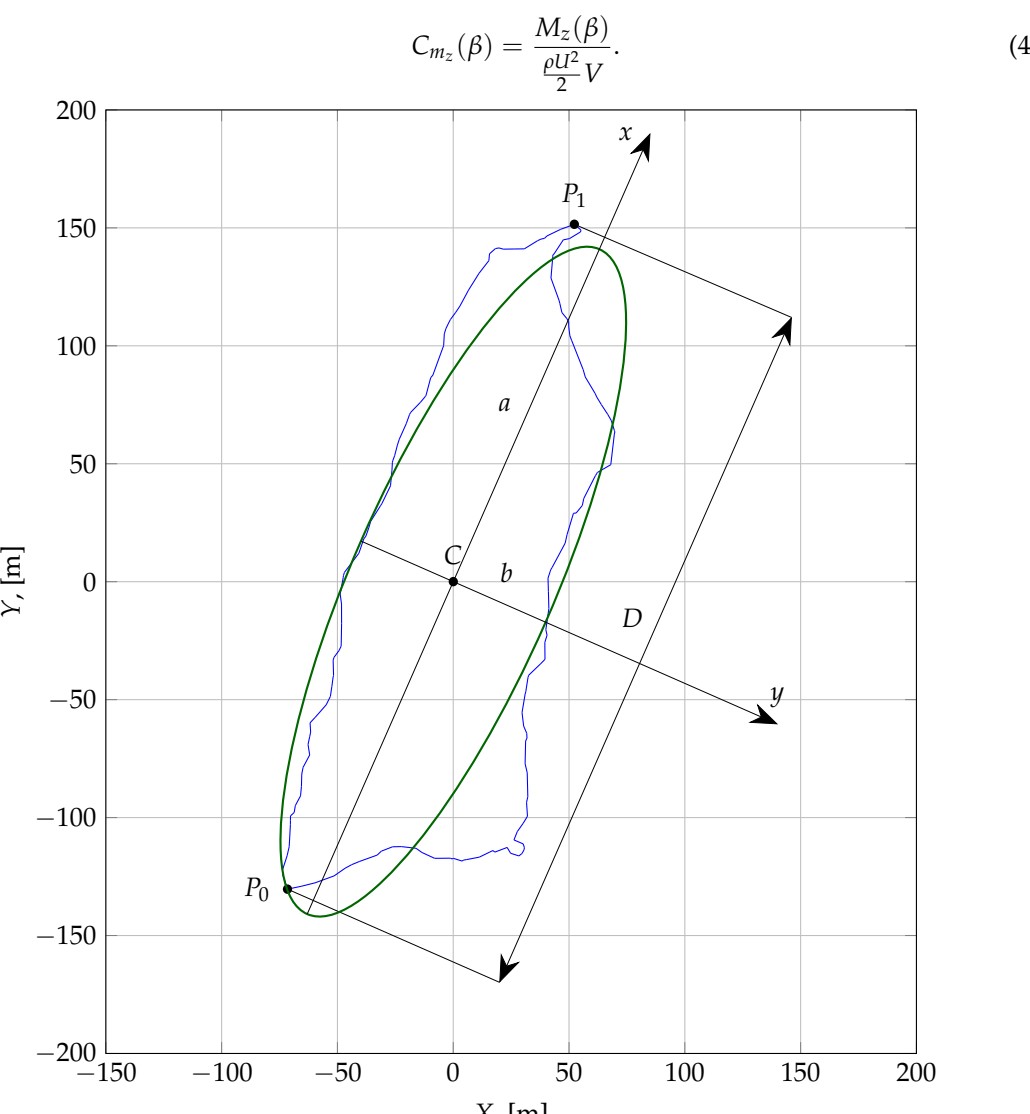

**Figure 1.** The waterline contour approximation of iceberg no. 1961, with an ellipse.

### 2.2. Reynolds-Averaged Navier–Stokes Approach and Closure Problem

To solve the problem of turbulent flow over an equivalent iceberg the OpenFOAM package was used, which uses mathematical modelling based on a system of Reynolds-averaged Navier–Stokes Equations (URANS) [19]:

$$\begin{cases} \dfrac{\partial U_i}{\partial x_j} = 0, \\[2mm] \dfrac{\partial U_i}{\partial t} + U_j \dfrac{\partial U_i}{\partial x_j} = -\dfrac{1}{\rho}\dfrac{\partial p}{\partial x_i} + \dfrac{\partial}{\partial x_j}\left( \nu \dfrac{\partial U_i}{\partial x_j} - \overline{u_i' u_j'} \right), \end{cases}\tag{5}$$

where $u_i = U_i + u_i'$ are the fluid particle velocities in the $i$ direction, $U_i$ are the mean velocities, $u_i'$ are the fluctuating velocities, $\overline{u_i' u_j'}$ are the second-order correlations, $\rho$ the density, $\nu$ the kinematic viscosity, $t$ the time, and $x_i$ the Cartesian coordinates.

According to J. Boussinesq's hypothesis, the Reynolds stresses are modelled as:

$$-\rho \overline{u_i' u_j'} = 2\mu_t S_{ij} - \frac{2}{3}\rho k \delta_{ij},\tag{6}$$

where $S_{ij} = \frac{1}{2}\left(\frac{\partial U_j}{\partial x_i} + \frac{\partial U_i}{\partial x_j}\right)$ is the mean rate of strain tensor, $k = \frac{1}{2}\overline{u'_i u'_i}$ is the turbulence kinetic energy, $\mu_t = \rho \nu_t$ is the turbulent viscosity, $\nu_t$ is the turbulent kinematic viscosity and $\delta_{ij}$ is the Kronecker symbol.

To close the set of Equations (5) and (6), the model of the shear stress transfer (MSST) [20] is used, well proven when applied to typical near-wall flows, including those with flow separation [21–23]. This is a generalization of two well-known turbulence models in the practice of engineering calculations: $k - \epsilon$ model [24] and low Reynolds $k - \omega$ Suffman–Wilcox model [25] (involved in the near-wall region). The model equations are [26]:

$$\nu_t = \frac{\alpha_1 \, k}{max(\alpha_1 \omega, SF_2)},$$

$$\frac{\partial k}{\partial t} + U_j \frac{\partial k}{\partial x_j} = P_k - \beta^* k\omega + \frac{\partial}{\partial x_j}\left[(\nu + \sigma_k \nu_t)\frac{\partial k}{\partial x_j}\right],$$

$$\frac{\partial \omega}{\partial t} + U_j \frac{\partial \omega}{\partial x_j} = \alpha S^2 - \beta\omega^2 + \frac{\partial}{\partial x_j}\left[(\nu + \sigma_\omega \nu_t)\frac{\partial \omega}{\partial x_j}\right] + 2(1 - F_1)\sigma_{\omega 2}\frac{1}{\omega}\frac{\partial k}{\partial x_i}\frac{\partial \omega}{\partial x_i},$$

where

$$P_k = \min\left(\tau_{ij}\frac{\partial U_i}{\partial x_j},\ 10\beta^* k\omega\right),$$

$$\sigma_k = \frac{F_1}{\sigma_{k1}} + \frac{1 - F_1}{\sigma_{k2}},\ \sigma_\omega = \frac{F_1}{\sigma_{\omega_1}} + \frac{1 - F_1}{\sigma_2},$$

$$F_1 = \tanh\left[\left[\min\left(\max\left(\frac{\sqrt{k}}{\beta^* \omega y},\ \frac{500\nu}{y^2\omega}\right),\ \frac{4\sigma_{\omega 2}k}{CD_{k\omega}y^2}\right)\right]^4\right],$$

$$CD_{k\omega} = \max\left(2\rho\sigma_{\omega 2}\frac{1}{\omega}\frac{\partial k}{\partial x_i}\frac{\partial \omega}{\partial x_i},\ 10^{-10}\right),$$

$$F_2 = \tanh\left[\left[\max\left(\frac{2\sqrt{k}}{\beta^* \omega y},\ \frac{500\nu}{y^2\omega}\right)\right]^2\right],$$

$$\phi = \phi_1 F_1 + \phi_2(1 - F_1),$$

with the coefficients $\alpha = 1, \beta = 0.072, \alpha_1 = \frac{5}{9}, \alpha_2 = 0.44, \beta_1 = 0.075, \beta_2 = 0.0828, \beta^* = 0.09,$ $\sigma_{k1} = 0.85, \sigma_{k2} = 1, \sigma_{\omega_1} = 0.85, \sigma_{\omega_2} = 0.856, \beta^*_\infty = 0.09$, $y$ as the distance to the nearest wall, $\phi = \{\alpha_\infty,\ \beta_i\}$ where the function $\phi_1$ represents the constants with the index "1" and $\phi_2$ represents the constants with the index "2", and $\alpha_{\infty,1} = \frac{\beta_1}{\beta^*} - \frac{\kappa^2}{\sigma_{\omega_1}\sqrt{\beta^*}}, \alpha_{\infty,2} = \frac{\beta_2}{\beta^*} - \frac{\kappa^2}{\sigma_{\omega,2}\sqrt{\beta^*}},$ $\kappa = 0.41$ is the T. von Karman's constant. Here, $\omega$ is the specific rate of dissipation (of the turbulence kinetic energy $k$ into internal thermal energy, to be exact), $P_k$ is the volumetric production rate of $k$, $\sigma_k$ and $\sigma_\omega$ are the the Prandtl numbers, and $F_1$ and $F_2$ are the blending functions.

At the initial moment of time, it is assumed that the dynamic system is in a rest state. At the input area of the outer boundary of the computational domain, the parameters of the undisturbed flow are set as $U = U_0, \frac{\partial p}{\partial \tilde{n}} = 0, \frac{\partial k}{\partial \tilde{n}} = 0$, and $\frac{\partial \omega}{\partial \tilde{n}} = 0$. The turbulence characteristics are formulated in the same way as in [15]. Thus, the turbulence energy at the inlet boundary $k_\infty$ is given by the corresponding degree of turbulence of the oncoming flow $Tu_\infty = 0.1\%$, and the scale of turbulence $L_\infty$ is chosen in the same order as the characteristic linear size of the object.

Concerning the outlet area of the outer boundary, soft boundary conditions (conditions for the continuation of the solution) are set as $\frac{\partial U}{\partial \bar{n}} = 0$, $p = p_0$, $\frac{\partial k}{\partial \bar{n}} = 0$, and $\frac{\partial \omega}{\partial \bar{n}} = 0$, and on the surface of the body no-slip conditions are imposed $U_x = U_y = U_z = 0$, $\frac{\partial p}{\partial \bar{n}} = 0$, $\frac{\partial k}{\partial \bar{n}} = 0$, and $\frac{\partial \omega}{\partial \bar{n}} = 0$, as shown in Figure 2a.

Wall functions were used in the near-wall area:

$$k_p = \frac{u_\tau^2}{\sqrt{C_\mu}}, \quad \omega_p = \frac{\sqrt{k_p}}{\sqrt[4]{C_\mu}\kappa y_p},$$

where $k_p$ and $\omega_p$ are values of turbulence kinetic energy and specific dissipation in the first knot, respectively, $y_p$ is the distance from the wall to the first knot, and $C_\mu = 0.09$ is a constant.

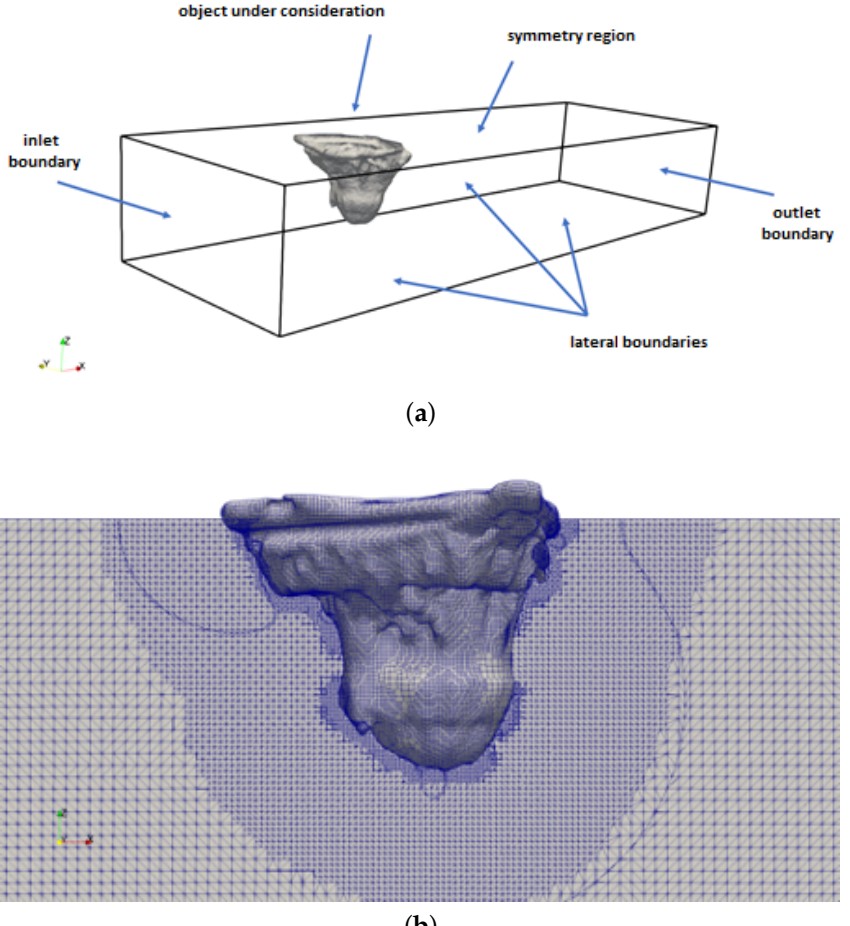

(**a**)

(**b**)

**Figure 2.** An example for iceberg no. 15 numerical simulation settings; (**a**) the boundary conditions; (**b**) the computational mesh for iceberg no. 15.

To automatically generate a computational mesh with a reduction in cells sizes to the object's surface in automatic mode, a special createMesh utility developed at the Laboratory of Applied Hydrodynamics of the St. Petersburg State Marine Technical University was used. The SnappyHexMesh utility from the OpenFOAM package was used to cut and approximate the object's surface from the initial mesh. An example of the computational mesh generated with help of the createMesh utility for iceberg no. 15 is given in Figure 2b. It was assumed that $y^+$ is in the range $30 \leq y^+ < 300$ [27] during mesh generation.

The Reynolds number was $Re = \frac{UL}{\nu} = 10^7$, as in this case the flow over an iceberg in this range of Reynolds numbers corresponds to a self-similar regime [28]. More details on the CFD simulations of icebergs can be found in [15].

*2.3. Results of CFD Simulations of the Flow over an Elliptical Cylinder and Half-Ellipsoid*

In the present investigation we used icebergs which were researched in details in previous articles [16,28,29]. Below, we briefly describe their general parameters. The essential parameters of the computed elliptical cylinders are given in Table 1.

**Table 1.** Parameters of the elliptical cylinders.

| Parameter | Elliptical Cylinders | | | |
|---|---|---|---|---|
| Length-to-beam ratio | 1.5 | 2.0 | 2.5 | 3.0 |
| Length, m | 75 | 100 | 125 | 150 |
| Beam, m | 50 | 50 | 50 | 50 |
| Draught, m | 124.3 | 93.2 | 74.6 | 62.2 |
| Volume displacement, m$^3$ | | 366,159 | | |
| Characteristic area, m$^2$ | | 5118.15 | | |
| Characteristic linear size, m | | 71.54 | | |

The essential parameters of the computed half-ellipsoids are given in Table 2.

**Table 2.** Parameters of half-ellipsoids.

| Parameter | Ellipsoids | | | |
|---|---|---|---|---|
| Length-to-beam ratio | 1.5 | 2.0 | 2.5 | 3.0 |
| Length, m | 75 | 100 | 125 | 150 |
| Beam, m | 50 | 50 | 50 | 50 |
| Draught, m | 186.8 | 140.1 | 113 | 93.4 |
| Volume displacement, m$^3$ | | 1,467,080.5 | | |
| Characteristic area, m$^2$ | | 12,972.5 | | |
| Characteristic linear size, m | | 113.1 | | |

In Figure 3, the streamlines around the elliptical cylinders with various $L/B$ ratios are demonstrated. Velocity vectors and relative excess pressures field around the elliptical cylinders when towed along the minor axis at the drift angle $\beta = 45°$ are shown in Figure 4. When the cylinders are towed, very intensive vortices are obtained in the trace, which is expected for bluff bodies, including icebergs. In this case, separation resistance significantly exceeds the friction resistance, which could be an indirect confirmation of the possibility of replacing an iceberg with an equivalent elliptical cylinder or ellipsoid. Here, the scale effect is not significant and the resistance coefficient weakly depends on the Reynolds number.

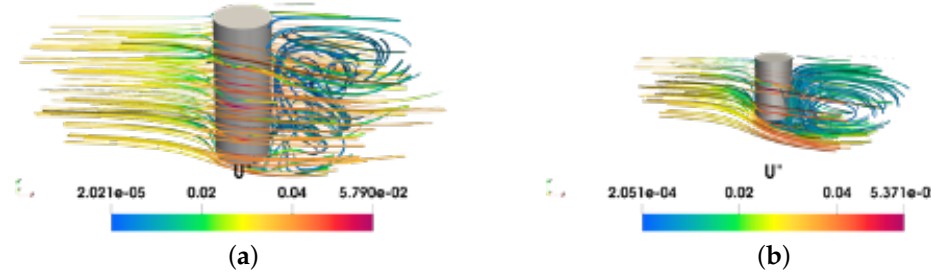

(**a**)　　　　　　　　　　　　　　　　　　　(**b**)

**Figure 3.** *Cont.*

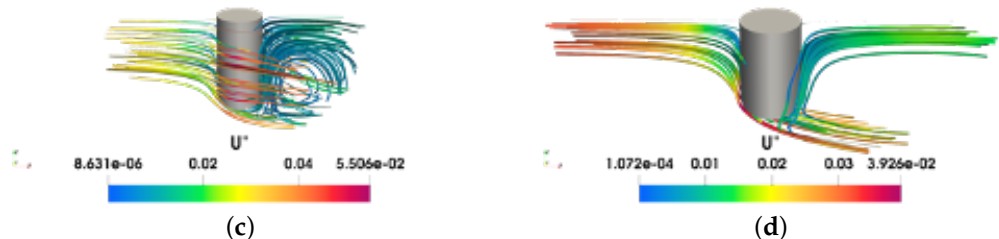

**(c)**                                      **(d)**

**Figure 3.** Streamlines around the elliptical cylinders when towed along the minor axis (**a**) $L/B = 1.5$; (**b**) $L/B = 2.0$; (**c**) $L/B = 2.5$; and (**d**) $L/B = 3.0$.

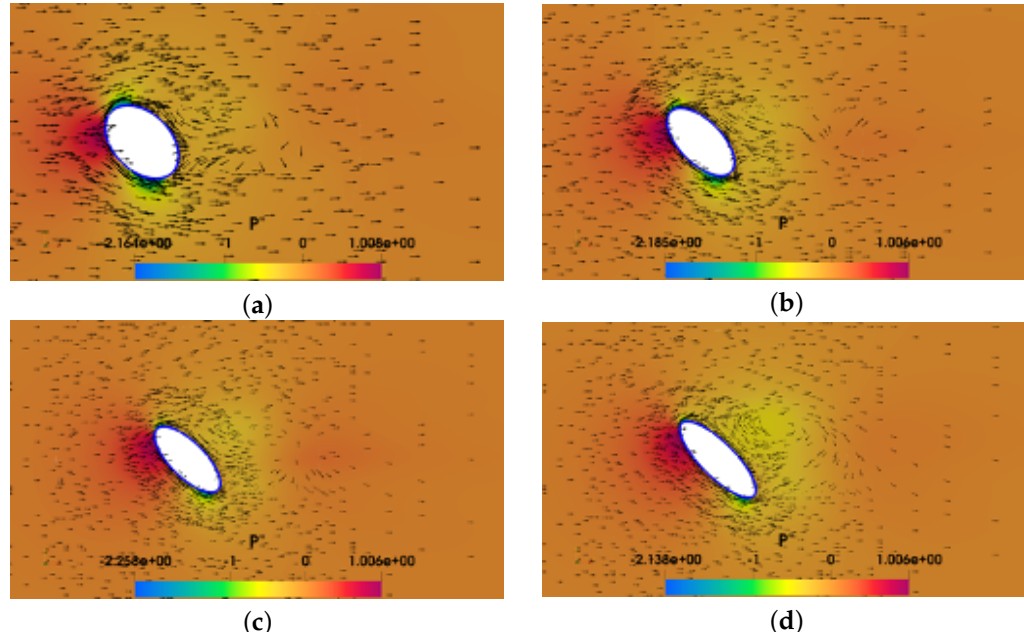

**Figure 4.** The velocity vectors and relative excess pressure field around the elliptical cylinders when towed along the minor axis at a drift angle $\beta = 45°$ (**a**) $L/B = 1.5$; (**b**) $L/B = 2.0$; (**c**) $L/B = 2.5$; and (**d**) $L/B = 3.0$.

The results of the equivalent cylinder hydrodynamic coefficient calculations and turning moment depending on the drift angle are shown in Figure 5. The expression (1) was used for the resistance and normal forces computation, while the relation (4) was used for the turning moment computation.

The results of the half-ellipsoids hydrodynamic coefficient computations are shown in Figure 6, considering the parameters given in Table 2 and the dependence on the drift angle.

The longitudinal coefficient $C_x$ of the cylinders has the maximal values at the drift angles of about $20°$, though its behaviour significantly depends on the $L/B$ ratio at large values of angle $\beta$. For large $L/B$, the longitudinal coefficient does not have a zero value, and is reasonably quite small. Yet, the normal force coefficient $C_y$ of such cylinders in this area is larger.

A significant difference in the behaviour of the longitudinal and normal force coefficient curves for different $L/B$ ratios is explained by the various ways of the vortex formation in the flow behind them.

At the same time, it is opposite for the longitudinal coefficients $C_x$ of ellipsoids: for small $L/B$ the curves do not cross 0, and for $L/B = 3.0$ there is a gap in the resistance at $45°$. The graph of the normal force coefficient qualitatively resembles that of a cylinder, but has significantly smaller values.

The moment coefficients of the cylinders is about two times larger for the ellipsoids and qualitatively very similar: in the both cases the moment coefficient value increases when the $L/B$ ratio is also increasing.

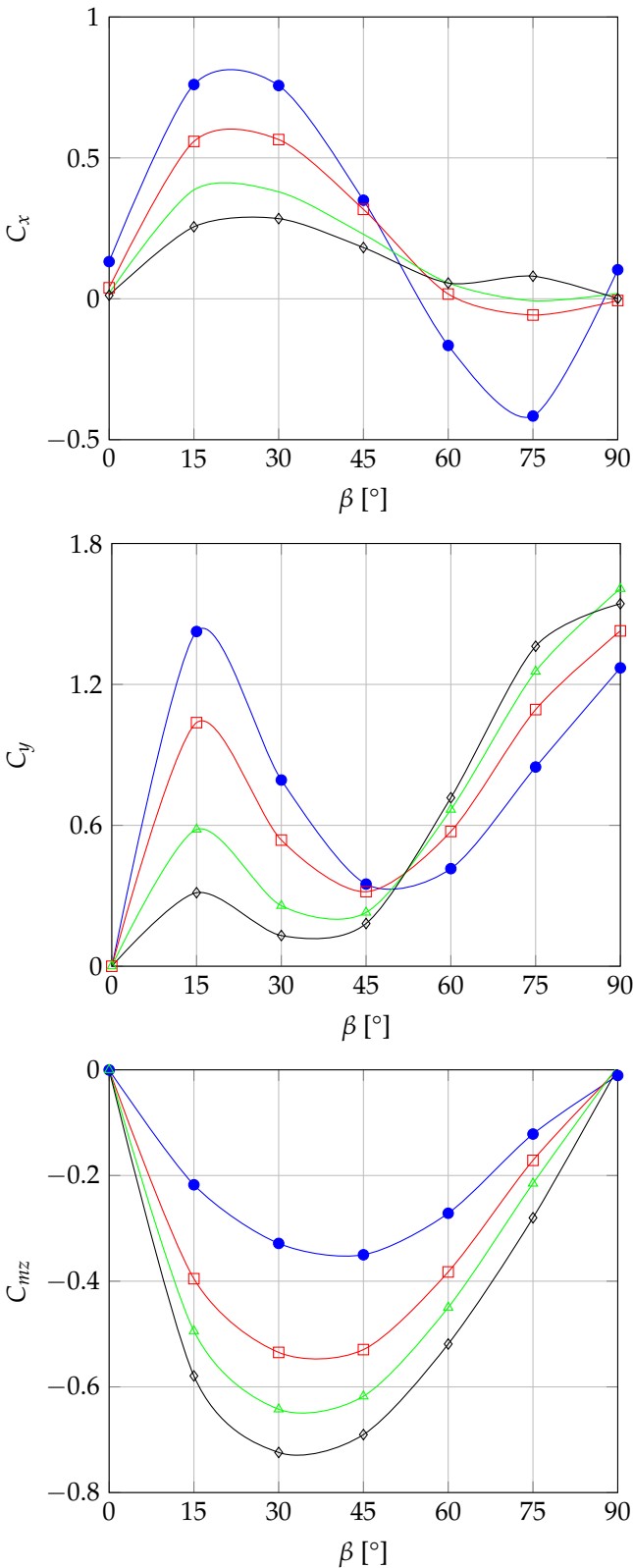

**Figure 5.** Hydrodynamic coefficients of cylindrical icebergs with various aspect ratios: (the blue line with circles is $L/B = 1.5$, the red line with squares is $L/B = 2.0$, the green line with triangles is $L/B = 2.5$, and the black line with diamonds is $L/B = 3.0$).

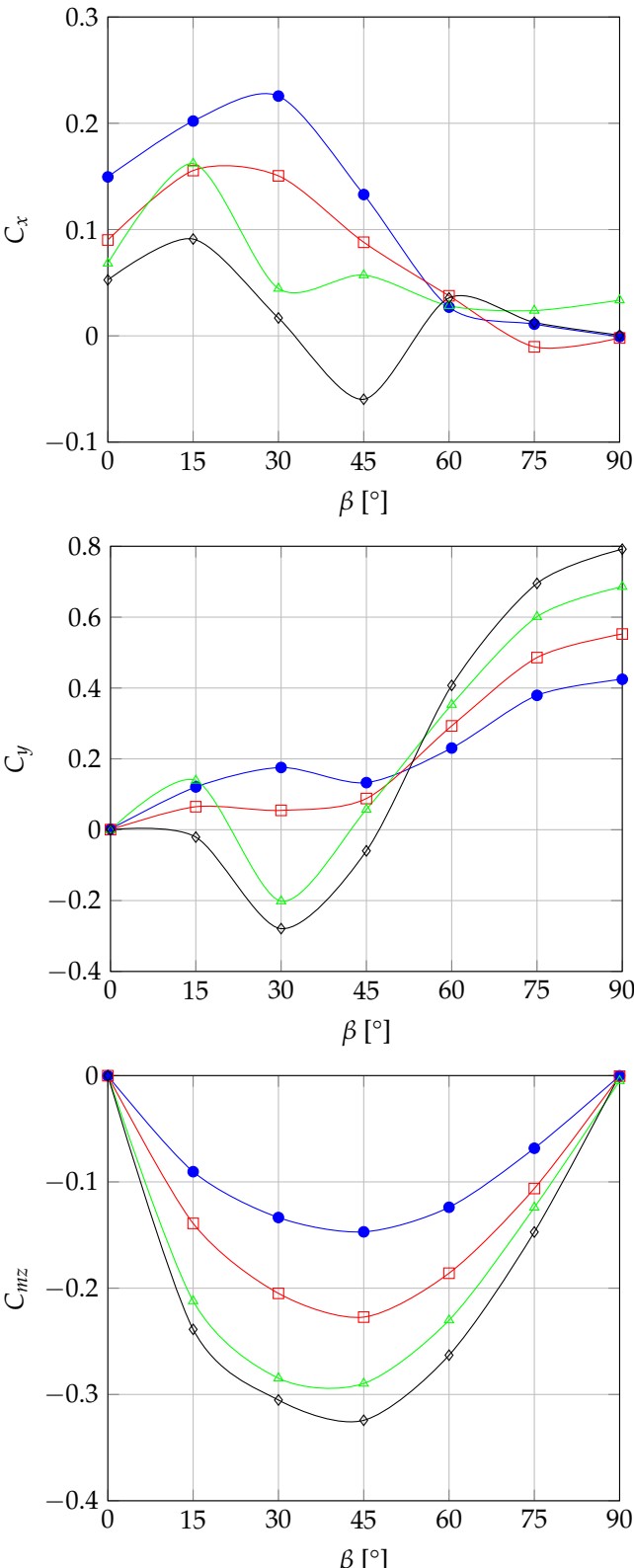

**Figure 6.** Hydrodynamic coefficients of half-ellipsoid icebergs with various aspect ratios: (the blue line with circles is $L/B = 1.5$, the red line with squares is $L/B = 2.0$, the green line with triangles is $L/B = 2.5$, the black line with diamonds is $L/B = 3.0$).

*2.4. Hydrodynamic Forces of Equivalent Icebergs*

An approximation of the computational results above allows the expressions for the hydrodynamic coefficients of an equivalent iceberg to be obtained. These approximations can be represented in a polynomial form:

$$C_x(\beta) = \left( a_x^4 \left( \frac{L}{B} \right)^4 + a_x^3 \left( \frac{L}{B} \right)^3 + a_x^2 \left( \frac{L}{B} \right)^2 + a_x^1 \left( \frac{L}{B} \right) + a_x^0 \right) Cos(\beta),$$

$$C_y(\beta) = \left( a_y^4 \left( \frac{L}{B} \right)^4 + a_y^3 \left( \frac{L}{B} \right)^3 + a_y^2 \left( \frac{L}{B} \right)^2 + a_y^1 \left( \frac{L}{B} \right) + a_y^0 \right) Sin(\beta), \quad (7)$$

$$C_m(\beta) = \left( a_m^4 \left( \frac{L}{B} \right)^4 + a_m^3 \left( \frac{L}{B} \right)^3 + a_m^2 \left( \frac{L}{B} \right)^2 + a_m^1 \left( \frac{L}{B} \right) + a_m^0 \right) Sin(2\beta),$$

where $a_x^i$, $a_y^i$ and $a_m^i$ are the coefficients of the approximation; $i = 1, 2, 3, 4$; $L$ is the length of the equivalent iceberg; and $B$ is the beam of the equivalent iceberg.

Coefficients of the approximations are given in Table 3.

**Table 3.** Coefficients of the approximations.

| Value | $a^4$ | $a^3$ | $a^2$ | $a^1$ | $a^0$ |
|---|---|---|---|---|---|
| | | Cylindrical iceberg | | | |
| $C_x$ | 0.110 | 0.085 | 0.145 | −0.84 | 1.3 |
| $C_y$ | 0 | −0.075 | −0.055 | 0.365 | 0.475 |
| $C_m$ | 0 | −0.131 | 0.955 | −2.485 | 1.655 |
| | | Elliptical iceberg | | | |
| $C_x$ | 0 | 0 | 0.0855 | −0.4603 | 0.6269 |
| $C_y$ | 0 | 0 | −0.25 | 1.335 | −0.2055 |
| $C_m$ | 0 | 0 | −0.21 | 1.1476 | −0.8795 |

The rotational part of the moment coefficient for an elliptical cylinder with relation $\frac{B}{T} = 2$ can be found using the formula:

$$C_m(\omega) = -0.0305 \left( \frac{L}{B} \right)^2 + 0.4011 \left( \frac{L}{B} \right)^2 - 2.175. \quad (8)$$

The rotational part of the turning moment in a dimensional form is as follows:

$$M_z(\omega) = k_{B/T} C_m(\omega) \omega |\omega| \rho \frac{L^4 T}{64}, \quad (9)$$

where $k_{B/T} = -0.04 \left( \frac{B}{T} \right)^3 + 0.5207 \left( \frac{B}{T} \right)^2 - 2.2968 \left( \frac{B}{T} \right) + 3.836$ is the coefficient, considering the deviation of given $B/T$ starting at 2.

When we obtain the accelerated motion parameters of an iceberg, the inertial forces must be taken into account. For instance, when we need to simulate the unsteady motion of an iceberg being towed by a tug vessel, we must consider its added (or virtual) masses, which can be found as [30]:

$$\lambda_{11} = \frac{4}{3} \pi \rho abc \frac{A_0}{2 - A_0}, \quad \lambda_{22} = \frac{4}{3} \pi \rho abc \frac{B_0}{2 - B_0}, \quad \lambda_{33} = \frac{4}{3} \pi \rho abc \frac{C_0}{2 - C_0},$$

$$\lambda_{44} = \frac{4}{15} \pi \rho \frac{abc (b^2 - c^2)^2 (C_0 - B_0)}{2 (b^2 - c^2) + (B_0 - C_0)(b^2 + c^2)},$$

$$\lambda_{55} = \frac{4}{15} \pi \rho \frac{abc (c^2 - a^2)^2 (A_0 - C_0)}{2 (c^2 - a^2) + (C_0 - A_0)(c^2 + a^2)},$$

$$\lambda_{66} = \frac{4}{15}\pi\rho \frac{abc\left(a^2 - b^2\right)^2 (B_0 - A_0)}{2(a^2 - b^2) + (A_0 - B_0)(a^2 + b^2)},$$

where

$$A_0 = abc \int_0^\infty \frac{du}{(a^2 + u)\sqrt{(a^2 + u)(b^2 + u)(c^2 + u)}},$$

$$B_0 = abc \int_0^\infty \frac{du}{(b^2 + u)\sqrt{(a^2 + u)(b^2 + u)(c^2 + u)}},$$

$$C_0 = abc \int_0^\infty \frac{du}{(c^2 + u)\sqrt{(a^2 + u)(b^2 + u)(c^2 + u)}},$$

In the initial stages of an investigation, it is possible to determine the added masses using the expressions for added masses of plane figures. For instance, for an ellipse with a semi-axes $a$ and $b$ and area $S = \pi ab$, the added mass coefficients can be determined as:

$$\lambda_{11} = \pi\rho b^2, \quad \lambda_{22} = \pi\rho a^2, \quad \lambda_{66} = \pi\rho \frac{a^2 - b^2}{8}.$$

For a flat plate with a chord equal to $2a$, the added mass coefficient can be determined as:

$$\lambda_{22} = \pi\rho a^2, \quad \lambda_{66} = \pi\rho \frac{a^4}{8}.$$

In uncommon cases, the added mass coefficients of flat figures are intermediate: their values are in between the range of the added mass coefficients for an ellipse and a flat plate.

Aerodynamic parameters of flat and 3D objects can either be found in the referenced literature or computed with the help of specialized modelling software.

As a result, we obtain the hydrodynamic model of a random iceberg, allowing us to determine its resistance, normal force and turning moment on the basis of substituting the iceberg with the equivalent elliptical cylinder or half-ellipsoid.

## 3. Comparison of the Hydrodynamic Parameters of Icebergs Computed with the Help of the Simplified Method and CFD Results

The suggested approach was proven by a comparison of the computed hydrodynamic parameters with the CFD simulations results obtained at the St. Petersburg State Marine Technical University (SMTU) and described in [15]. With this purpose, four real icebergs shapes were selected, having the most typical underwater shapes: iceberg no. 1961 is a prism with a trapezoid at the base, iceberg no. 1960 is a prism with a rectangle at the base, iceberg no. 124 is elongated in the vertical direction and iceberg no. 15 is non-tabulated. These icebergs were selected from a list of 36 icebergs, investigated in the Arctic region from 2016 to 2017. Surveying the underwater (by sonar) and surface (by helicopter) parts of these icebergs was carried out and saved as a clouds of points. The authors created 3D models of the four selected icebergs with the help of a special self-created software, allowing the cloud of points to be converted into IGES files.. The primary characteristics of the icebergs are given in Table 4 (location of the centre of gravity was counted from the waterline).

To investigate the grid independence, nine computational grids with 1 to 18 millions cells were created and the numerical simulation of the flow over an iceberg for $Re = 10^7$ was conducted. The analysis indicates that the resistance coefficient has a constant value in the range of $1.545 \leq y^+ \leq 263$. In this case, there were approximately 6,000,000 cells. The total drag coefficient $C_d$ depending on $y^+$ can be found in Figure 7. An example of the CFD simulation results obtained is given in Figure 8. More details on the CFD simulations of icebergs can be found in [15]. Icebergs with equivalent ellipsoids are shown in Figure 9.

**Table 4.** Iceberg parameters.

| Parameter | Iceberg No. 15 | Iceberg No. 124 | Iceberg No. 1960 | Iceberg No. 1961 |
|---|---|---|---|---|
| Length, m | 148 | 39.0 | 72.6 | 158 |
| Beam, m | 148 | 31.1 | 76.4 | 301 |
| Hight, m | 25.93 | 5.25 | 12.433 | 13.808 |
| Draught, m | 114 | 46.5 | 51.0 | 59.9 |
| Volume, m$^3$ | 854,565 | 26,713 | 157,483 | 1,575,309 |
| Volume displacement, m$^3$ | 764,435 | 23,895 | 140,874 | 1,409,162 |
| Overwater volume, m$^3$ | 90,130 | 2818 | 16,609 | 166,147 |
| Characteristic linear size, m | 95 | 30 | 54 | 116 |
| Characteristic area, m$^2$ | 9005 | 894 | 2916 | 13,539 |
| Waterline area, m$^2$ | 12,840.666 | 833.84 | 2913.696 | 21,065.82 |
| Center of gravity, m | −34.0 | −16.6 | −20.2 | −24.5 |

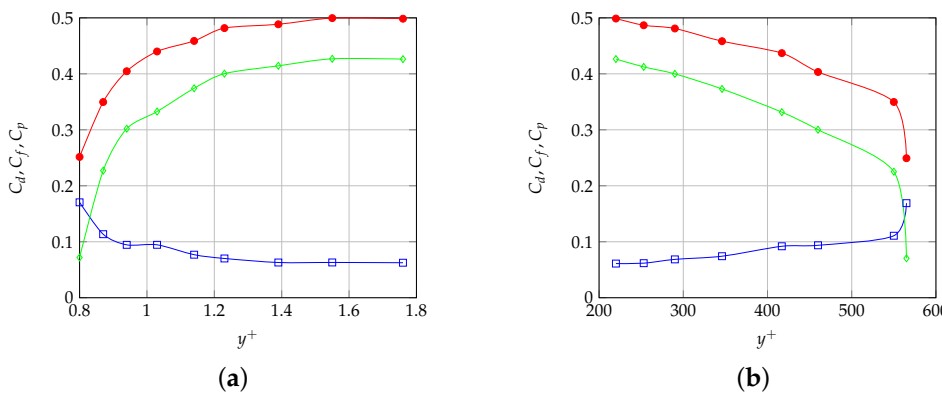

(a)　　　　　　　　　　　　(b)

**Figure 7.** Drag coefficient of iceberg no. 15 depending on $y^+$; (**a**) the area of minimal $y^+$ values; (**b**) area of maximal $y^+$ values of the drag coefficient $C_d$ according to (2), friction part of drag coefficient $C_f$, and viscous pressures part of the drag coefficient $C_p$.

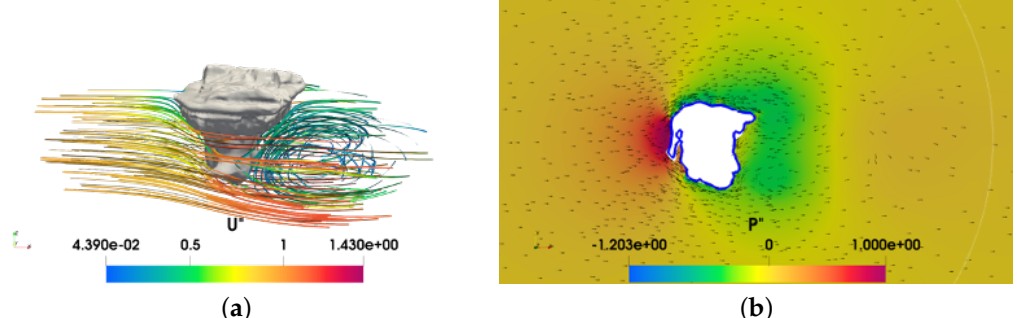

(a)　　　　　　　　　　　　(b)

**Figure 8.** An example of the CFD simulation of the flow over iceberg no. 15, $\beta = 270°$; (**a**) streamlines over the underwater part; (**b**) velocity field in the waterplane area.

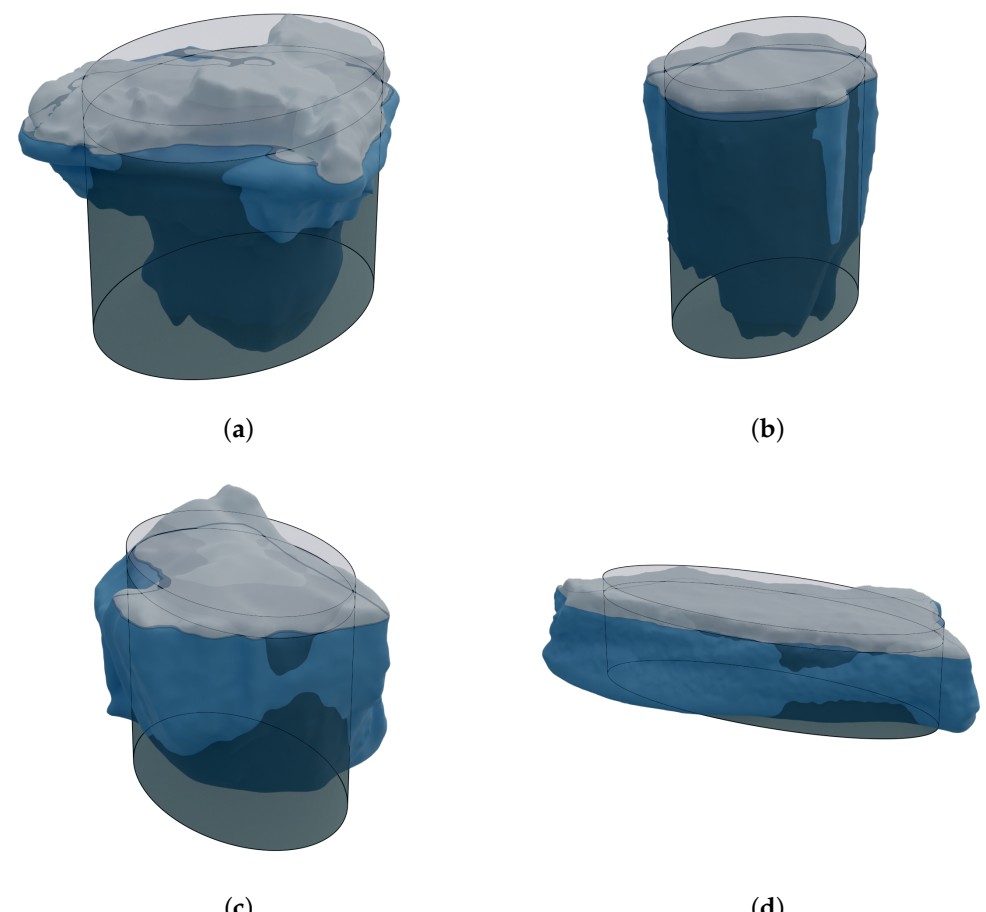

(**a**)

(**b**)

(**c**)

(**d**)

**Figure 9.** Considered icebergs and their equivalent ellipsoids; (**a**) iceberg no. 15, (**b**) iceberg no. 124, (**c**) iceberg no. 1960, and (**d**) iceberg no. 1961.

A comparison of the obtained approximations with corresponding simulations are shown in Figures 10–13.

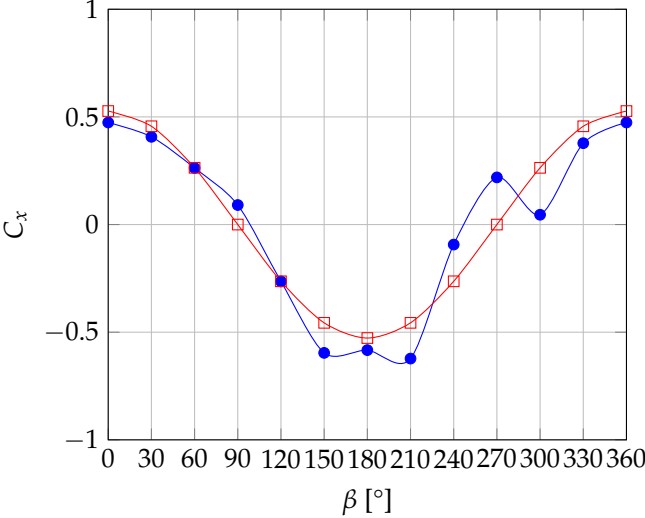

**Figure 10.** *Cont.*

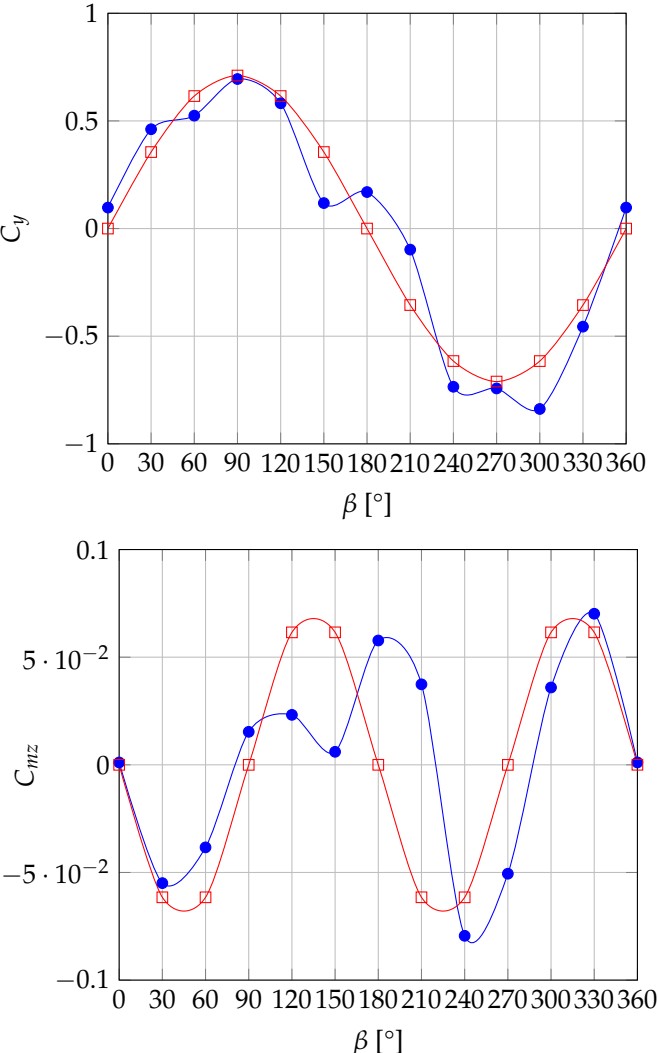

**Figure 10.** Comparison of the hydrodynamic coefficients of iceberg no. 15. (blue lines with circles are measurements, and red lines with squares are approximations).

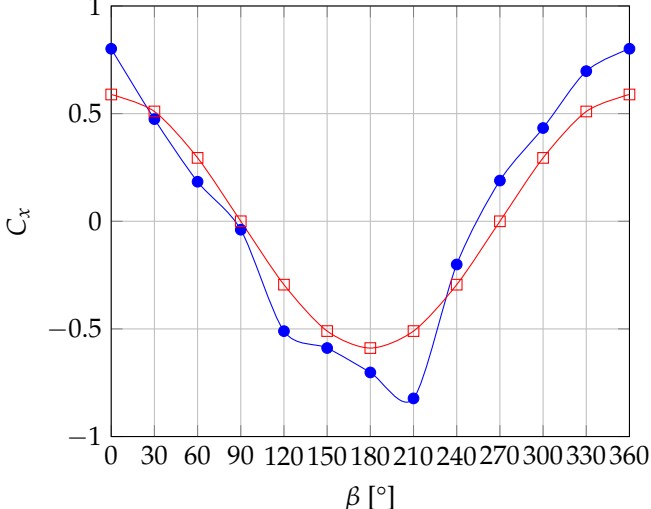

**Figure 11.** *Cont.*

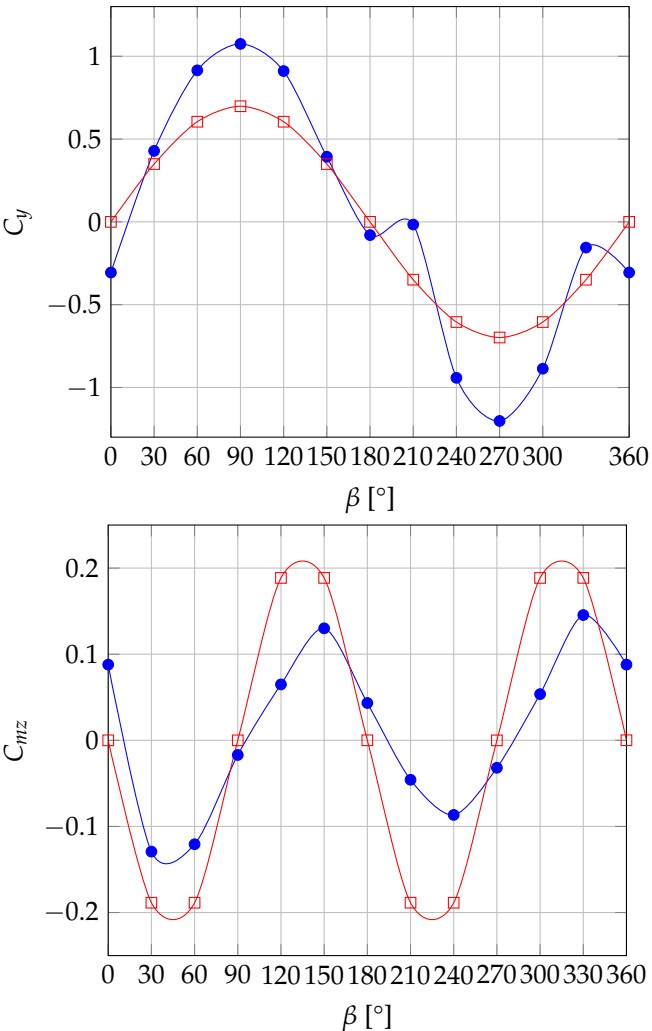

**Figure 11.** Comparison of the hydrodynamic coefficients of iceberg no. 124. (blue lines with circles are measurements, and red lines with squares are approximations).

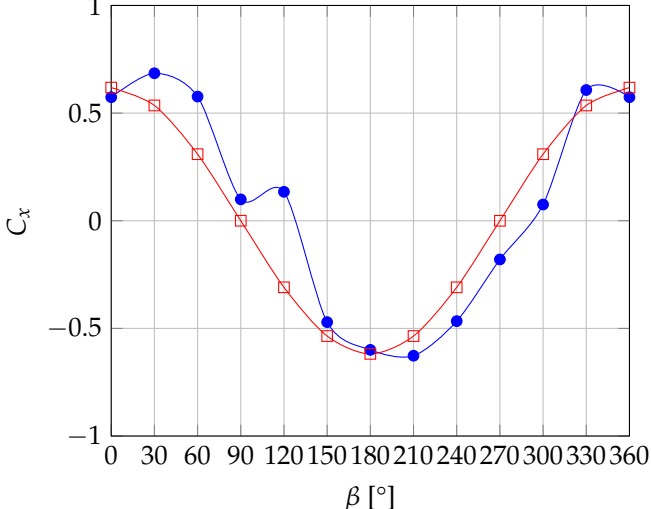

**Figure 12.** *Cont.*

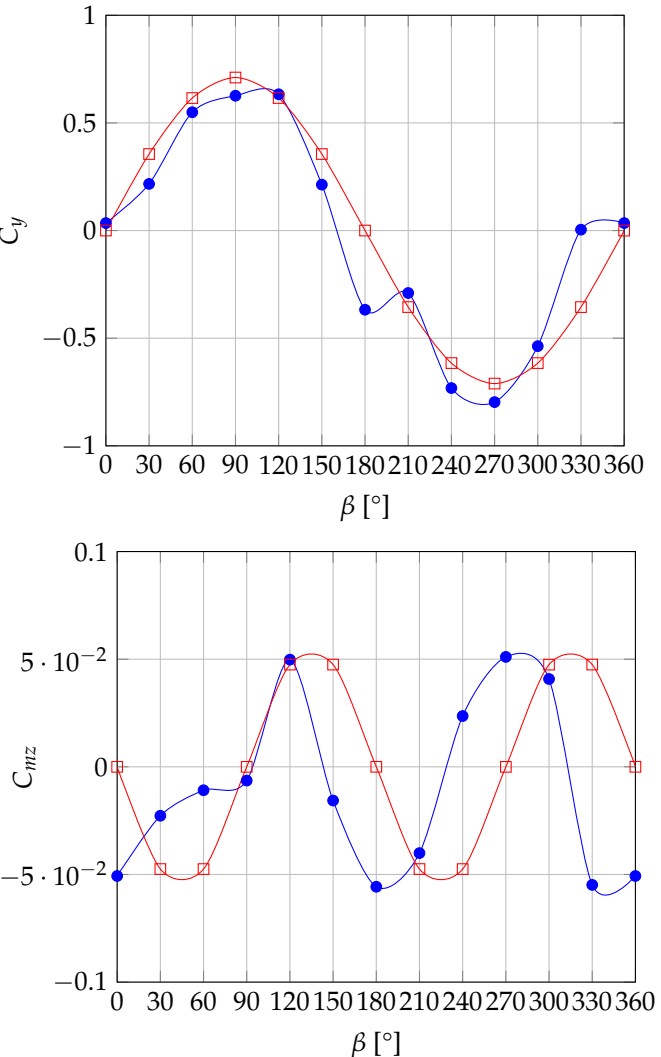

**Figure 12.** Comparison of the hydrodynamic coefficients of iceberg no. 1960. (blue lines with circles are measurements, and red lines with squares are approximations).

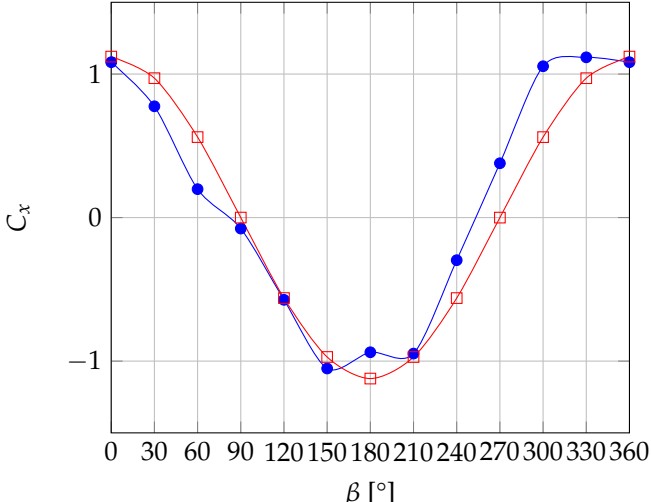

**Figure 13.** *Cont.*

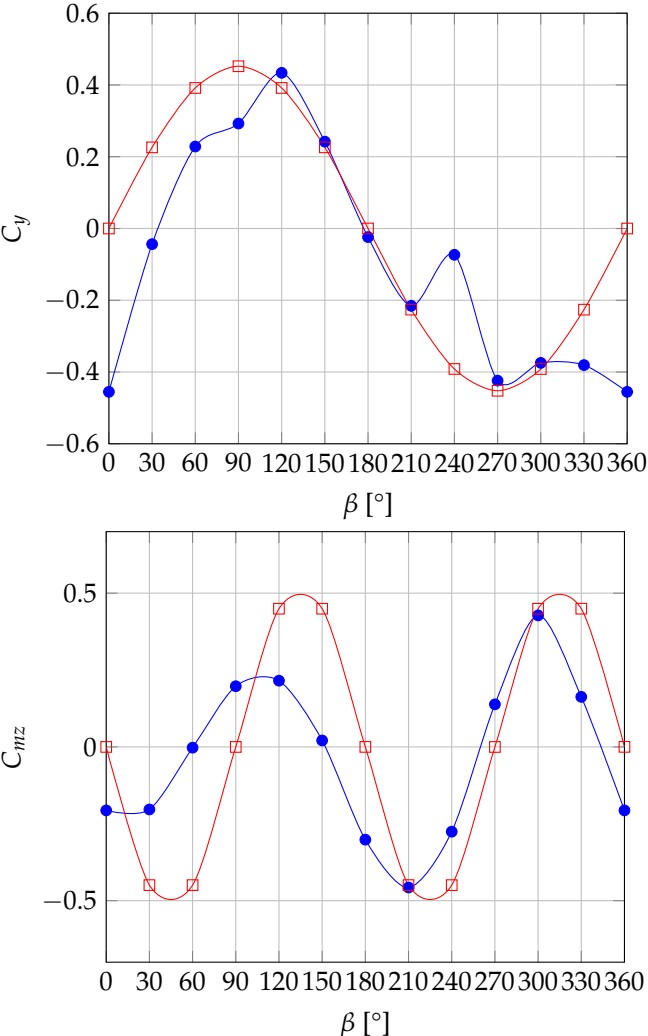

**Figure 13.** Comparison of the hydrodynamic coefficients of iceberg no. 1961. (blue lines with circles are measurements, and red lines with squares are approximations).

For iceberg no. 15, the resistance at the drift angle of 0° is overestimated (by approximately 30%) when compared to the simulation data, and at 180° it is very close to it (an error of approximately 5%). The normal force is qualitatively approximated close to the simulation data, having an error of 2% and 15% in peak values at 90° and 270°, respectively. However, by the employed approximation, an equivalent iceberg cannot consider the asymmetry of a real iceberg, which is not crucial for the purposes of the present study. The torque curve has a drift angle shift of approximately 40°, while the peak values are in good agreement with the simulated ones.

For iceberg no. 124 (a fragment-shaped iceberg), the qualitative data is close to the simulated one. However, the peak values of the force coefficients are significantly underestimated (up to 35%), while the moments, on the contrary, are overestimated (up to 45%).

The same is applied to iceberg no. 1960 as to iceberg no. 15. Despite a number of deviations of the approximated curve from the simulated one, these icebergs are qualitatively close. Peak values differ slightly from each other (resistance differs by 1.4%, normal force differs by 12.3%, and the moment differs by 7.0%), allowing us to conclude that the proposed approximations are adequate.

For iceberg no. 1961, the approximations give significant deviations for both the normal force curve and moment. Particularly noteworthy are the zones 0–90 and 300–360. The used approach shows a qualitative discrepancy with the simulations for the latter zone. This suggests that either the simulation was performed poorly, or the underwater

part differs significantly from the elliptical cylinder. However, the amplitude values for the normal force differ by about 5%, and by less than 2% for the moment, allowing us to conclude that the proposed approach is applicable for this iceberg as well.

Figure 14 demonstrates the drag coefficient, determined according to Formula (2). The figure clearly shows that the approximation and test results are in satisfactory agreement with each other for all icebergs, except for iceberg no. 124. Whereas the calculated curve for iceberg no. 124 is significantly lower than the simulated one, with an error of 42% at a drift angle of 270°. By this reason, we can also conclude that the required method may be used with limitations for icebergs of this such type.

Principally, we can confirm that the proposed method gives qualitatively similar results for iceberg no. 15, no. 1960 and no. 1961 in comparison with the CFD simulations. the vertical moment coefficient deviations are related to the fact that the underwater shape of real icebergs is not an elliptical cylinder and has irregular characteristics. On the other hand, the method allows a sufficient evaluation of its peak values to be obtained for practical usage. Such evaluations are also sufficient for obtaining primary estimates of their behaviour. The longitudinal force is qualitatively approximated quite well, and so is the transverse force. The moments show some phase shift; however, the amplitudes are close to the experimental ones, which means that the orientation of the iceberg model during towing will differ from the real one. However, the required thrust on the hook will correspond to its value in real cases of icebergs towing and this provides the iceberg trajectory prediction by the proposed method.

At the same time, approximations for the fragment-shaped iceberg no. 124 give significant deviations both for the normal force curve (by about 40% at a drift angle of 90°) and for the moment (by about 100% at a drift angle of 240°). Therefore, the simplified approach must be applied with caution for this type of iceberg. However, in the considered case, there is good qualitative agreement between the approximation and simulation results. These approximations, nevertheless, can be used in the rough prediction of forces and moments in the initial stages of making decisions. Yet, it is necessary to note that the table-shaped iceberg no. 124 is characterized by an anomalous relationship between the area of the waterline and the draft, and it also has low stability. Such objects can be only be observed in the fjords of the Arctic islands (where iceberg no. 124 was found during the observation). Producing glaciers descend into those fjords. Thus, this vertically elongated iceberg is anomalous for the Arctic seas, and the flow around it is more complicated than that of most tabular icebergs (e.g., icebergs no. 1961 and no. 1960). Therefore, the proposed method can be used for the hydrodynamic parameter prediction of regular icebergs with underwater shapes close to the studied icebergs no. 15, no. 1960 and no. 1961.

Thus, we can say that the proposed approximation has good qualitative and, in some cases, quantitative agreement, which indicates the possibility of applying this approach in practice.

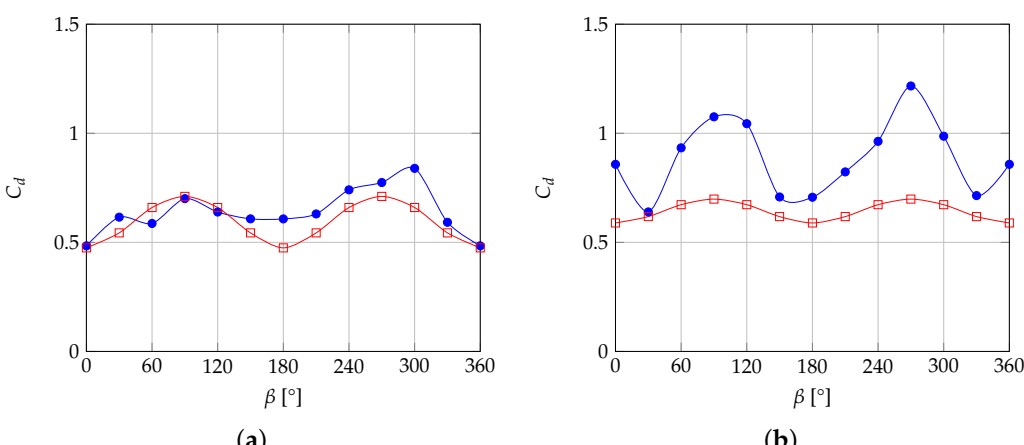

(a)                    (b)

**Figure 14.** *Cont.*

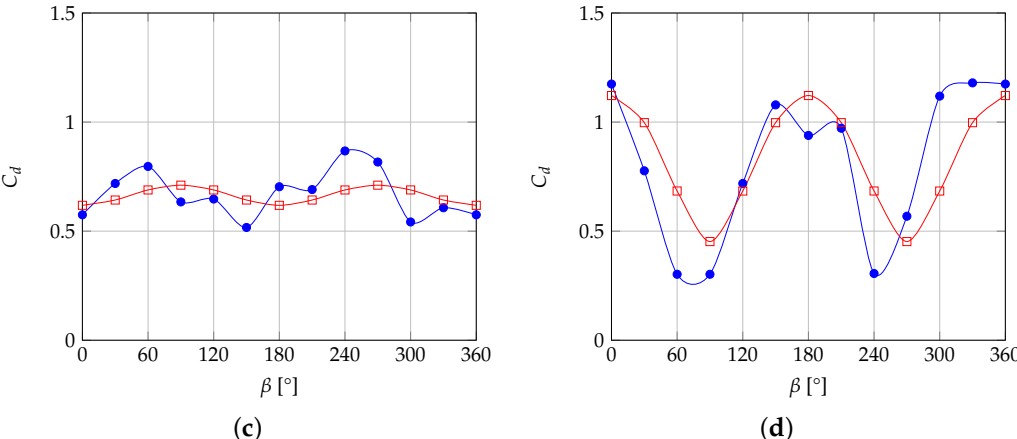

**Figure 14.** Full drag force coefficient $C_d$ (as a function of the drift angle $\beta$) of: (**a**) iceberg no. 15, (**b**) iceberg no. 124, (**c**) iceberg no. 1960, and (**d**) iceberg no. 1961. Blue lines with circles are measurements, and red lines with squares are approximations.

## 4. Conclusions

A new approach to iceberg hydrodynamic parameter prediction was proposed in the present article. This new approach may be used in computer simulations and decision support systems for cases when information on an iceberg's underwater shape is insufficient or absent.

This approach is based on the assumption that an iceberg waterplane can be approximated by an ellipse, which has the biggest axis equal to the maximal length of the iceberg and the second is perpendicular to it, as shown in Figure 1. This allows an iceberg's underwater shape to be represented as an equivalent ellipsoid or elliptical cylinder based on an ellipse and a set of hydrodynamic parameter approximations as functions of the length-to-beam ratio and drift angle, suggested in the Equations (7)–(9). This allow us to compute the hydrodynamic forces and moment of an iceberg during its motion on a curvilinear trajectory.

Comparison of the obtained hydrodynamic parameters of equivalent icebergs with model trials of real icebergs shows that the proposed approach allows to quantitatively calculate the close force coefficient curves. The comparison shows that for three out of the four investigated full-scale icebergs, the longitudinal and normal forces can be found for use in simulation systems for iceberg dynamics prediction. At the same time, the turning moment coefficient can be quantitatively determined with sufficient accuracy. Yet, there is a phase shift until 70° (for iceberg no. 1961), which should be considered in practical use of the suggested approach.

A noticeable drawback of the approach is the two-plane symmetry of equivalent ellipsoid or elliptical cylinders. This leads to a much more predictable iceberg trajectory than expected: the underwater shape of an iceberg is neither ellipsoid nor cylindrical [16]. Despite this, the suggested approach allows the forecasting of the iceberg's location area in its free drift, or may be used to simulate iceberg towing.

We have not discussed iceberg aerodynamic forces in this article, as the key objective of the present research was iceberg towing, but they may also be found using the proposed approach.

The presented approach may be improved in the future when more information about full-scale iceberg hydrodynamics is available. Furthermore, more iceberg parameter deviations need to be included in the suggested approximations.

In addition to the coefficients of hydrodynamic resistance and added masses, the metacentric height and frequency response of icebergs are important parameters for iceberg towing. The considered question on whether it is possible to replace the real shape of the underwater part of the iceberg with an elliptical cylinder or an ellipsoid to ensure compliance with these parameters needs additional research.

**Author Contributions:** Conceptualization, K.K., Y.E. and D.N.; methodology, A.M.; validation, E.N.; formal analysis, K.K.; investigation, A.M. and N.T.; software, A.S.; resources, D.N.; data curation, D.N. and K.K.; software A.S.; writing—original draft preparation, D.N. and A.S.; writing—review and editing, D.N. and E.N; visualization, A.S. and Y.E.; supervision, D.N.; project administration, D.N.; funding acquisition, K.K. and Y.E. All authors have read and agreed to the published version of the manuscript.

**Funding:** This research was partially funded by the Ministry of Science and Higher Education of the Russian Federation as a part of the World-Class Research Center program: Advanced Digital Technologies (contract No. 075-15-2022-312 dated 20 April 2022).

**Institutional Review Board Statement:** Not applicable.

**Informed Consent Statement:** Not applicable.

**Data Availability Statement:** Not applicable.

**Conflicts of Interest:** The authors declare no conflict of interest.

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
