# Peer review of "A Simplified Method of Iceberg Hydrodynamic Parameter Prediction"

_water, doi:10.3390/w15101843_

Round 1
Reviewer 1 Report
This is an interested topic for water. It can be published after major reversion. Followings are the modification suggestions.
1. Reference support should be provided for the drag coefficient 1.2 mentioned in lines 47 to 51, as well as for the cylindrical experiments that match it.
2. It is recommended to put the application section 71-80 later. Because the main work and unique advantages of this paper are described in lines 61-70, and the structure of this article is presented in lines 81-86.
3. Please simplify the lines 81-117 and avoid complete duplication with the abstract part. The background in the introduction is sufficient. This part only needs to introduce some parameters required for this simplified method of icebergs used in this paper, as well as their sources.
4. It is recommended to replace the drag force letters X and Y in the x and y directions defined in formula (1). They are duplicated with the horizontal and vertical coordinates in Figure 1, which can easily lead to misunderstandings.
5. Have other turbulence models been compared with SST models in these cases? And the explanation should be given for the letters used in line 151, such as turbulent kinetic energy, dissipation rate, etc.
6. Have grid independence checks or grid impact analysis been conducted? It is recommended to add a clear numerical model diagram, including grid division, boundary conditions, etc. In addition, reference [15] was not found, please provide the paper.
7. In section 2.3, please add some description and analysis of CFD results.
8. Compared to cylindrical structures, more icebergs are closer to conical shapes in the underwater part, as shown in Figure 5(a) and (b). Why not consider approximating an iceberg as an elliptical cone structure?
9. What are the possible reasons for the deviation between the simplified approximate method and measurement in Figures 6-9? Especially for the Cmz curve, there are obvious differences in peak value and their positions.
10. Figure 10 lacks the vertical coordinate title.
11. Lines 258-259 suggest that this method is not suitable for icebergs of the No.124. What features of No.124 make it not applicable for this method? Moreover, I suggest that you choose an iceberg with a shape similar to No. 124 and compare its approximate and experimental values.
Author Response
Dear Sir or Madam,
Thank you very much for year warm revision. Please find below answres on your remarks.
1. Reference support should be provided for the drag coefficient 1.2 mentioned in lines 47 to 51, as well as for the cylindrical experiments that match it.
Answer: There is a reference [1] in the submitted manuscript («Allen, J. Iceberg Study, Saglek, Labrador; Cruise Report CSS Dawson, August 7-August 26, 1972 Technical report, Memorial University of Newfoundland, Faculty of Engineering and Applied Science report, 1972.»). A comparison with a cylinder was made in the cited article.
2. It is recommended to put the application section 71-80 later. Because the main work and unique advantages of this paper are described in lines 61-70, and the structure of this article is presented in lines 81-86.
Answer: This has been rewritten and updated in the latest version of the manuscript (lines 71-80 have been moved to the very beginning of the Section 2).
3. Please simplify the lines 81-117 and avoid complete duplication with the abstract part. The background in the introduction is sufficient. This part only needs to introduce some parameters required for this simplified method of icebergs used in this paper, as well as their sources.
Answer: This has been rewritten and updated in the latest version of the manuscript.
4. It is recommended to replace the drag force letters X and Y in the x and y directions defined in formula (1). They are duplicated with the horizontal and vertical coordinates in Figure 1, which can easily lead to misunderstandings.
Answer: This has been fixed in the latest version of the manuscript. Thank you
5. Have other turbulence models been compared with SST models in these cases? And the explanation should be given for the letters used in line 151, such as turbulent kinetic energy, dissipation rate, etc.
Answer: In the present investigation we did not use another turbulence models. Our experience shows that this model allows to obtain adequate simulation results for such objects. Some explanations of letters added.
6. Have grid independence checks or grid impact analysis been conducted? It is recommended to add a clear numerical model diagram, including grid division, boundary conditions, etc. In addition, reference [15] was not found, please provide the paper.
Answer: Yes, the grid independency was studied. We have updated the manuscript with this information.
The reference link is https://www.isope.org/wp-content/uploads/2022/06/abst-32-2-p143-ik09-Tryaskin.pdf
We must confess that we have made a mistake during our work under the manuscript - we used for comparison CFD simulation data instead of measurements. CFD is very close to measurements (see attached example for the iceberg 1961: blue line is towing tank test, red line is CFD, green line is wind tunnel tests), but not the same. We also have found a mistake in the moment coefficient of iceberg 1961 - during conversion into the connected reference frames it got the incorrect sign. Sorry for that.
7. In section 2.3, please add some description and analysis of CFD results.
Answer: It has been added in the latest version of the manuscript.
8. Compared to cylindrical structures, more icebergs are closer to conical shapes in the underwater part, as shown in Figure 5(a) and (b). Why not consider approximating an iceberg as an elliptical cone structure?
Answer: Icebergs no.124, no. 1961, and no. 1960 are table-shaped, whereas iceberg no. 15 is non-table-shaped. Table-shaped icebergs are a rectangular prism, at the base of which lies either a trapezoid or a rectangle. Therefore, the approximation of table-shaped icebergs by means of an elliptical cylinder is quite justified. As for non-table-shaped icebergs, it is practically impossible to determine the shape of their underwater part from the known above-water part without sonar surveying. Authors propose to approximate their underwater shape with an elliptical cylinder, while, as follows from the analysis, such an approximation reflects the hydromechanical characteristics of the underwater part with sufficient accuracy for practical application. Thus, by means of an elliptical cylinder, it is possible to simulate (or to model, to be exact, in terms of hydromechanics) the underwater part of both table-like and non-table-shaped icebergs.
9. What are the possible reasons for the deviation between the simplified approximate method and measurement in Figures 6-9? Especially for the Cmz curve, there are obvious differences in peak value and their positions.
Answer: The vertical moment coefficient deviations is connected with the fact that underwater shape of real icebergs is not the elliptical cylinder and has irregular character in reality. But we suppose the method allows obtaining enough close evaluation of its peak values, which can be successfully used for icebergs hydrodynamics prediction in simulation software.
10. Figure 10 lacks the vertical coordinate title.
Answer: Figure 10 has been updated, so that it now has «Cd» vertical axis name. We supposed that if all diagrams at the Figure shows the same values it is enough indicate it in the caption
11. Lines 258-259 suggest that this method is not suitable for icebergs of the No.124. What features of No.124 make it not applicable for this method? Moreover, I suggest that you choose an iceberg with a shape similar to No. 124 and compare its approximate and experimental values.
Answer: Table-shaped iceberg no. 124 was characterized by an anomalous relationship between the area of the waterline and draft, and it also had low stability. Such objects can be observed only in the fjords of the Arctic islands. Producing glaciers descend into those fjords. Thus, this vertically elongated iceberg is anomalous for the Arctic seas, and its flow around is more complicated than that of most tabular icebergs (e.g. icebergs no. 1961 and no. 1960).
On the other hand the proposed method could be applicable to iceberg no. 124 also, but if in this case no. 15 and no. 1960 are approximated more or less well, then hydrodynamics of no. 1961 is predicted incorrectly both by forces and by moments. As we mentioned earlier, in our opinion no. 124 is not typical iceberg therefore we decided to calibrate our approach on the most typical icebergs, which in our case are 15, 1960 and 1961.
Moderate changes to English language have been added to enhance readability of the manuscript

Reviewer 2 Report
"A Simplified Method of An Iceberg Hydrodynamic Parameters Prediction" presents a method for predicting the hydrodynamic parameters of an iceberg by approximating the waterline as an ellipse. The proposed method is based on a combination of physical principles and empirical data. It uses a simplified model to estimate the key parameters that govern the motion of an iceberg, such as its drag coefficient, moment of inertia, and hydrostatic restoring force. There are several points that the authors should clarify:
1. Did the manuscript only focus on four icebergs because obtaining "full-scale" data is challenging, or were these four icebergs chosen because they are more suitable for the proposed model?
2. The authors proposed that their method yields similar qualitative results for icebergs 15, 1960, and 1961. However, it is evident from Figures 8 and 9 that there is a significant difference in the C_mz results. It would be beneficial if the authors could provide an explanation in the manuscript for the observed drift.
3. Line 269, "So that the approach must be applied with caution 269 for this iceberg." Can the authors provide guidance on the extent to which the model can be trusted and what its limitations are?
4. Line 291, the authors suggested "3 out of 4 investigated ... can be found with suffinicient accuracy...". However, there is inadequate support to validate the method. The sample size is too small, comprising only four samples, and there is no information on whether the sample is biased or not. I would recommend that the authors refrain from using the term "accurate" unless they can confirm the specific situations where this method can be reliably applied.
5. Section 3, the authors provided only the outcomes without investigating the underlying reasons for the diverse outcomes. Therefore, a more comprehensive analysis should be provided by the authors.
Author Response
Dear Sir or Madam,
Thank you very much for year warm revision. Please find below answres on your remarks.
1. Did the manuscript only focus on four icebergs because obtaining "full-scale" data is challenging, or were these four icebergs chosen because they are more suitable for the proposed model?
Answer: The following subsequent work has been conducted:
In 2016-2017 a series of experiments was carried out on towing 36 icebergs. At the same time, surveying of the underwater (sonar) and surface (helicopter) parts of some of these icebergs was carried out. 3D models were created, correspondingly. Thus, it was possible to obtain the shape of real icebergs.
In 2020, numerical and basin modeling was carried out to determine the hydromechanical characteristics of four of the studied icebergs. Choice of icebergs for the study origitaned from the following: we assumed that we should consider three characteristic table-shaped icebergs and one non table-shaped (iceberg no. 15, to be exact). Those three aforementioned table-shaped icebergs are iceberg no. 1961, iceberg no. 1960 and iceberg no. 124. Iceberg no. 1961 is a prism with the trapezium at the base weighing about 1 million tons. Iceberg no. 1960 is also a prism but with a rectangle at the base weighing about 100 thousand tons. Iceberg no. 124 is an elongated one in the vertical direction weighing about 15 000 tons.
Once data on four icebergs has been processed, the question arose on the possibility of approximating the underwater part by some geometric figure; so that, authors proposed an elliptical cylinder and prepared a rationale, which became the subject of the work under discussion.
2. The authors proposed that their method yields similar qualitative results for icebergs 15, 1960, and 1961. However, it is evident from Figures 8 and 9 that there is a significant difference in the C_mz results. It would be beneficial if the authors could provide an explanation in the manuscript for the observed drift.
Answer: The explanation is added
3. Line 269, "So that the approach must be applied with caution 269 for this iceberg." Can the authors provide guidance on the extent to which the model can be trusted and what its limitations are?
Answer: Authors claim that for table-like icebergs with a small length-to-draft ratio, the model somewhat underestimates the resistance, while qualitatively correctly describing the change in coefficients from the angle of attack. It should be noted that icebergs like iceberg no. 124 can only exist in calm water conditions (specifically, iceberg no. 124 was found in the fjord), since they have a low stability margin.
4. Line 291, the authors suggested "3 out of 4 investigated ... can be found with suffinicient accuracy...". However, there is inadequate support to validate the method. The sample size is too small, comprising only four samples, and there is no information on whether the sample is biased or not. I would recommend that the authors refrain from using the term "accurate" unless they can confirm the specific situations where this method can be reliably applied.
Answer: Authors agree with the reviewer’s comment, the terms «accurate», «accuracy» and the rest of that line have been deleted and are no longer used in the manuscript.
5. Section 3, the authors provided only the outcomes without investigating the underlying reasons for the diverse outcomes. Therefore, a more comprehensive analysis should be provided by the authors.
Answer: Some analisys has been added into the section 3.
We must confess that we have made a mistake during our work under the manuscript - we used for comparison CFD simulation data instead of measurements. CFD is very close to measurements (see attached example for the iceberg 1961: blue line is towing tank test, red line is CFD, green line is wind tunnel tests), but not the same. We also have found a mistake in the moment coefficient of iceberg 1961 - during conversion into the connected reference frames it got the incorrect sign. Sorry for that.

Round 2
Reviewer 1 Report
Thank you very much for adopting my suggestions and your revision. But I think you should be more careful. When I opened the reply, I saw two obvious word errors in the first sentence. It can be published after your minor modification. Here are my further comments.
1. Please merge the Figure 2 and Figure 3.
2. L/B= 1:1.5 or 1.5? why it is different? For example, in Figures 4, 5 and in the Table 1, etc.
3. Do lines 183-187 mean that the larger the L/B, the more severe the flow separation? You can refer to some paper about the circular cylinder in hydrodynamics. Additionally, you have conducted numerical simulations of two different structures. Why is the flow field of the ellipsoidal not shown in Figure 4 compared to it? After comparing these two structures, it would be more logical to choose the method of using cylindrical structures to simplify icebergs in Section 3.
4. In line 195, you should distinguish the Cx, Cy or Cd that you described. Just using the ‘resistance’ is not very clear.
5. From Figure 7(a), when y+ >0.945, Cd is not stable. How can you obtain a constant value in the range of 0.945 ≤ y+ ≤ 263?
Author Response
Please find below the answers. 1. Please merge the Figure 2 and Figure 3. Figures 2 and 3 have been merged. 2. L/B= 1:1.5 or 1.5? why it is different? For example, in Figures 4, 5 and in the Table 1, etc. This was a mistake, thank you. It has been fixed. 3. Do lines 183-187 mean that the larger the L/B, the more severe the flow separation? You can refer to some paper about the circular cylinder in hydrodynamics. Additionally, you have conducted numerical simulations of two different structures. Why is the flow field of the ellipsoidal not shown in Figure 4 compared to it? After comparing these two structures, it would be more logical to choose the method of using cylindrical structures to simplify icebergs in Section 3. The aforementioned means that the flow structure depends on L/B. If the L/B ratio is close to 1, we have streamlines picture in the middle of the cylinder, which could be compared with flow over an infinite cylinder, shown in Van Dyke's handbook (in our case, draught-to-length ratio will be 3.75). When L/B ratio is increasing, then in the limit we shall obtain flow over a flat plate set cross the flow. In this case, we shall see two coherent vortex structures at the edges of the plate. Now if we assume that two this edges move to each other (L/B is decreasing), then we shall obtain interaction of these two vortex structures, which leads to complexity of the flow in the cylinder trace (and cylinders have not fixed separation points). That is what we mean when match the computed results with bluff bodies. We have added a new figure 4 to illustrate velocity vectors and relative excess pressures field around the cylinders . By this reason, it is not so easy to find similar simulations, and we have made the computations by ourselves. There are many simulations, for instance, made in the study below https://skill-lync.com/student-projects/flow-over-a-cylinder-185 but these are not the same. As for streamlines around ellipsoids, we have learned the technical report but it does not contain these flow pictures. Unfortunately, the most of the simulations were made in 2020/21 years and not available at this time, sorry. As an option, we can exclude ellipsoids simulation results. 4. In line 195, you should distinguish the Cx, Cy or Cd that you described. Just using the ‘resistance’ is not very clear. Explanations have been added. 5. From Figure 7(a), when y+ >0.945, Cd is not stable. How can you obtain a constant value in the range of 0.945 ≤ y+ ≤ 263? This is mistake made during the manuscript preparation. For sure, the behavior is opposite: as it was mentioned in description, constant values of Cd were obtained in the range of 1 < y+ < 265. It has been fixed. Thank you very much.